# Consequences of climate-induced range expansions on multiple ecosystem functions

Jared A. Balik [1,2,3✉], Hamish S. Greig[2,4], Brad W. Taylor [1,2] & Scott A. Wissinger[2,3]

Climate-driven species range shifts and expansions are changing community composition, yet the functional consequences in natural systems are mostly unknown. By combining a 30-year survey of subalpine pond larval caddisfly assemblages with species-specific functional traits (nitrogen and phosphorus excretion, and detritus processing rates), we tested how three upslope range expansions affected species' relative contributions to caddisfly-driven nutrient supply and detritus processing. A subdominant resident species (*Ag. deflata*) consistently made large relative contributions to caddisfly-driven nitrogen supply throughout all range expansions, thus "regulating" the caddisfly-driven nitrogen supply. Whereas, phosphorus supply and detritus processing were regulated by the dominant resident species (*L. externus*) until the third range expansion (by *N. hostilis*). Since the third range expansion, *N. hostilis*'s relative contribution to caddisfly-driven phosphorus supply increased, displacing *L. externus*'s role in regulating caddisfly-driven phosphorus supply. Meanwhile, detritus processing contributions became similar among the dominant resident, subdominant residents, and range expanding species. Total ecosystem process rates did not change throughout any of the range expansions. Thus, shifts in species' relative functional roles may occur before shifts in total ecosystem process rates, and changes in species' functional roles may stabilize processes in ecosystems undergoing change.

[1] Department of Applied Ecology, North Carolina State University, Raleigh, NC 27695, USA. [2] Rocky Mountain Biological Laboratory, Crested Butte, CO 81224, USA. [3] Departments of Biology and Environmental Science, Allegheny College, Meadville, PA 16335, USA. [4] School of Biology and Ecology, University of Maine, Orono, ME 04469, USA. ✉email: Balikj3@gmail.com

Species range shifts or expansions occur throughout the historical biogeography of many native and introduced species[1,2], and are a well-documented adaptation to changing climatic conditions[3,4]. However, regardless of the causal mechanisms, species range shifts or expansions (hereafter, "range expansion") can change the species composition of recipient ecosystems[5]. This introduction of novel functional traits[6] and alteration of food web interactions[7] could alter species' contributions to ecosystem processes. Although short-term, small-scale experiments provide valuable insight towards functional outcomes of compositional shifts[8], predicting when and where species range expansions will occur remains a barrier to studying outcomes in natural systems[9]. Furthermore, outcomes of species range expansions can and likely will change over time as populations of range expanding and resident species fluctuate, but they are rarely studied across long timeframes that span multiple generations of researchers[10]. Similarly, ecosystems could receive multiple range expanding species over time, particularly where range expansions are prompted by climate warming[3]. Thus, long-term community composition data present unique opportunities to study how species range expansions, species losses, and other compositional shifts modulate species' contributions to ecosystem processes in natural systems.

One promising approach to predicting how compositional shifts could modulate species' direct effects on ecosystem processes is to combine long-term community composition data with species functional traits. Functional or effect traits are those that provide mechanistic links between organisms and ecosystem processes[11], often describing physiological characteristics that influence energy or material flows such as plant leaf and root stoichiometry or animal nutrient excretion rates[12]. Although interactions with other species and abiotic environmental factors also modulate overall ecosystem processes[13,14], scaling species' abundance and traits to ecosystem fluxes provides first-principles predictions of species' contributions[15–17].

Here, we apply this approach to a 30 year census of larval caddisflies (Trichoptera; annual generations) that dominate detritivore biomass in subalpine permanent ponds[18]. This census has documented three sequential upslope range expansions by caddisfly species *Limnephilus picturatus* in 1998, *Grammotaulius lorretae* in 2006, and *Nemotaulius hostilis* in 2016, as well as orders-of-magnitude changes in the abundance of common and subdominant resident caddisfly species (Fig. 1A, B). Including the range expanding species, this caddisfly assemblage has high interspecific trait variation in species-specific N and P excretion and detritus processing rates[19,20]. The influence of interspecific variation in invertebrate functional traits on fundamental ecosystem processes like nutrient cycling and detritus breakdown is increasingly recognized in aquatic[21,22] and terrestrial ecosystems[11]. In this system, animal-driven N and P supply contributions provide large proportions of N and P demand[23], and the caddisfly assemblage dominates coarse detritus breakdown[24]. Thus, we used key caddisfly functional traits and long-term abundance data to predict species-specific contributions to nutrient supply and detritus processing throughout the three range expansions.

Here, we focused on relative contributions of different caddisfly species to their assemblage's total contribution to N and P supply and detritus processing, rather than their contribution relative to the entire animal assemblage because our evidence for range shifts and long-term abundance data are limited to caddisflies. Our approach generates mechanistic predictions of species' relative contributions to ecosystem processes, such as N and P supplied by animals, that can be difficult to isolate in situ from other environmental factors such as nutrient uptake. Thus, our primary objective was to explore how the predicted relative contributions to multiple ecosystem processes of a dominant resident, a group of subdominant residents, and a group of range expanding species changed over time throughout three range expansions. We tested the hypothesis that successive range expansions would reduce the abundance of dominant resident *Limnephilus externus*[25], causing declines in its historically large relative contribution to the caddisfly assemblage's total nutrient supply and detritus processing (Fig. 1C). In contrast, we expected abundance of range expanding species to increase over time, along with their relative contributions to ecosystem processes. These predicted shifts in caddisfly abundance are consistent with strong intraguild interactions[18,26] and resource limitation[27] documented in this system. Next, our second objective was to explore how any changes in caddisfly assemblage evenness resulting from successive range expansions would influence redundancy in the assemblage's contributions to ecosystem processes. We expected that caddisfly assemblage evenness would increase through the addition of new species and numerical declines in the dominant resident (Fig. 1D). Greater caddisfly assemblage evenness was expected to increase redundancy in species' relative contributions to ecosystem processes[28]. Consequently, we tested for a negative relationship between evenness and aggregate variability of species' contributions to ecosystem processes (Fig. 1E). Specifically, aggregate variability of species' contributions to ecosystem processes includes among-pond variation in species' contributions and covariances among all species pairs.

## Results and discussion

### Caddisfly abundance and predicted ecosystem process contributions throughout sequential range expansions.
Many montane insect populations are declining due to climate-driven changes in temperature and precipitation patterns[29]. Systematic declines in caddisfly abundance have not been observed in the Rocky Mountains of western Colorado, USA, but since 1998 the assemblage has experienced the addition of three range expanding caddisfly species. Throughout these range expansions, caddisfly species' relative abundances have changed (e.g., significant species × year × range expansion term in Table S1; Fig. 2A, B), and consequently so have their relative contributions to ecosystem processes (Table S2; Fig. 2C–E). To evaluate the hypotheses for our primary objective, we examined trends in species abundance and relative contributions during each range expansion.

We expected the arrival of range expanding species to cause numerical declines in the dominant resident *L. externus* and thus reduce its contribution to ecosystem processes relative to range expanding species. However, this only occurred during one of three range expansions. During the first range expansion, by *L. picturatus*, the *L. externus* population increased by 1.3 individuals/m²/year after rapid recovery from a population crash that all residents experienced prior to the first range expansion (Fig. 2A; first expansion period *L. externus* abundance × time, slope linear contrast $p < 0.001$). There were no trends in the abundance of any other species during the first range expansion (linear contrasts $p > 0.05$). Similarly, there were no trends in species' abundances during the second range expansion by *G. lorretae* (linear contrasts $p > 0.05$). Consequently, there were no trends in the relative contribution of any caddisfly groups to ecosystem processes throughout the first or second range expansions (Fig. 2C–E; linear contrasts $p > 0.05$).

In contrast to the first and second range expansions, the third range expansion by *N. hostilis* significantly altered species abundances and relative contributions to ecosystem processes. Specifically, abundances of *L. externus* and subdominant resident *Asynarchus nigriculus* declined by ~1.9 individuals/m²/year

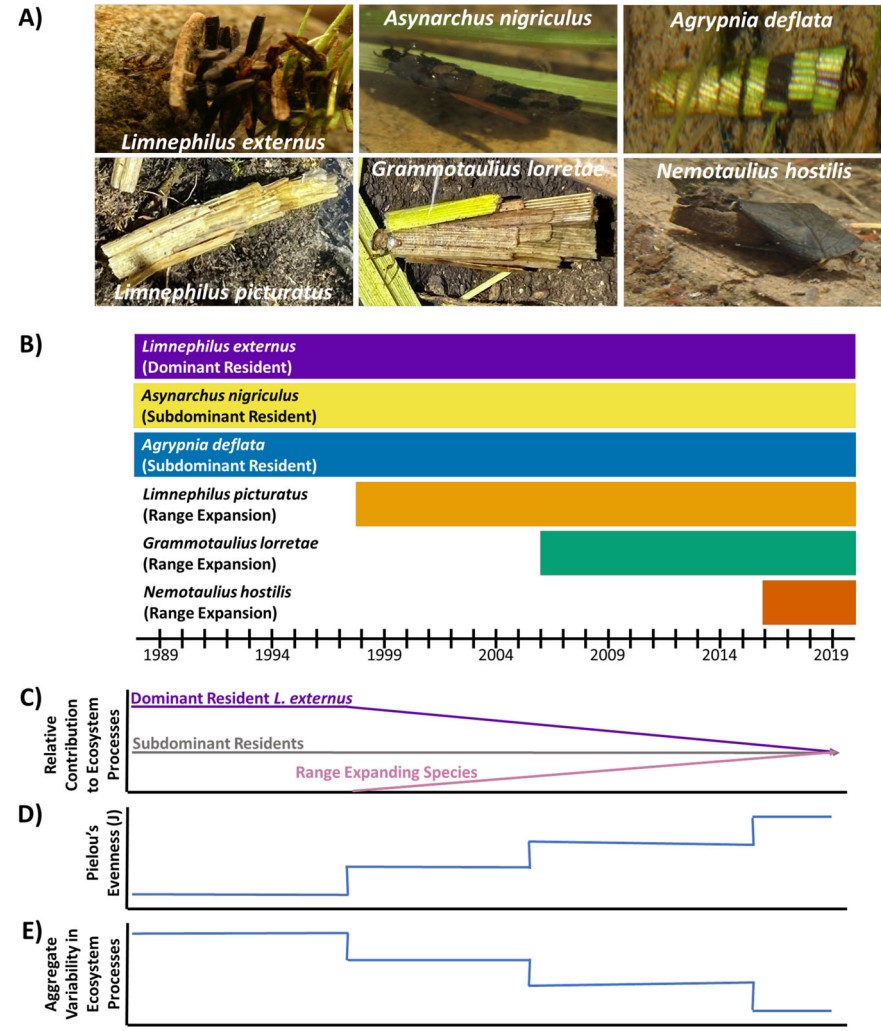

**Fig. 1 Conceptual diagram of caddisfly range expansions and hypothesized outcomes for caddisfly assemblage evenness, aggregate variability in ecosystem processes, and relative contributions to ecosystem processes of dominant and subdominant residents and range expanding species. A** Top row: photographs of larvae of resident caddisfly species; bottom row: photographs of larvae of range expanding caddisfly species. **B** Timeframes of the presence of each species at the Mexican Cut. Resident *L. externus* (numerically dominant caddisfly), *As. nigriculus*, and *Ag. deflata* were present since long-term surveys began in 1989. Range expanding *L. picturatus* arrived in 1998, followed by *G. lorretae* in 2006, and most recently *N. hostilis* in 2016. **C** We expected relative contributions of *L. externus* to ecosystem processes to decline following arrival of range expanding species because we expected these species to provide larger relative contributions to ecosystem processes over time. **D** We hypothesized that caddisfly assemblage evenness would increase as range-expanding species arrive at Mexican Cut and dominance of *L. externus* would decline. **E** Similarly, we hypothesized that greater caddisfly assemblage evenness over time would provide greater functional redundancy and thus reduce aggregate variability in the assemblage's contributions to ecosystem processes. Here, aggregate variability includes variability ecosystem process contributions among ponds and covariances among all species pairs.

(Fig. 2A; $p = 0.012$; $p = 0.054$) while *N. hostilis* increased by ~1.4 individuals/m²/year ($p < 0.001$) and exceeded the abundance of *L. externus* by 2018. Consequently, the relative contribution of *L. externus* to P supply and detritus processing declined by 11.2% and 13.9% annually (Fig. 2C, $p < 0.001$; 1E, $p < 0.001$) and the relative contribution of the group of range expanding species to P supply increased by 14.5% annually (Fig. 2C, $p < 0.001$). Despite declining *An. nigriculus* abundance, there was no trend in the group of subdominant resident's contribution to both ecosystem processes (P: $p = 0.917$; detritus: $p = 0.464$) which was consistently low (averaging <38% of P and <37% of detritus) compared to *L. externus*'s large contributions. Thus, the relative contributions of dominant resident and range expanding species trended in opposite directions during the third range expansion, and there were no trends in the relative contributions of subdominant species during any of the three range expansions.

The contrasting trends in relative contributions of the dominant resident *L. externus* and the range expanding species *N. hostilis* (Table S3; Fig. 2C, E) suggest that future changes in their contributions to ecosystem processes are possible should *L. externus* continue to be numerically replaced[10]. However, our estimates suggest that there have not been trends in the total contribution of the caddisfly assemblage to ecosystem processes over the last 30 years (Figure S1A–C). This could indicate that the total ecosystem process contributions of the caddisfly assemblage are constrained by energetic equivalence within the assemblage[30] or by the supply or quality of detrital resources[31]. Nonetheless, even if there were no trends in the total contribution of the assemblage, species relative functional roles could have changed[32]. For example, because relative contributions of *L. externus* were large and did not trend in any direction throughout the first or second range expansions, it effectively regulated

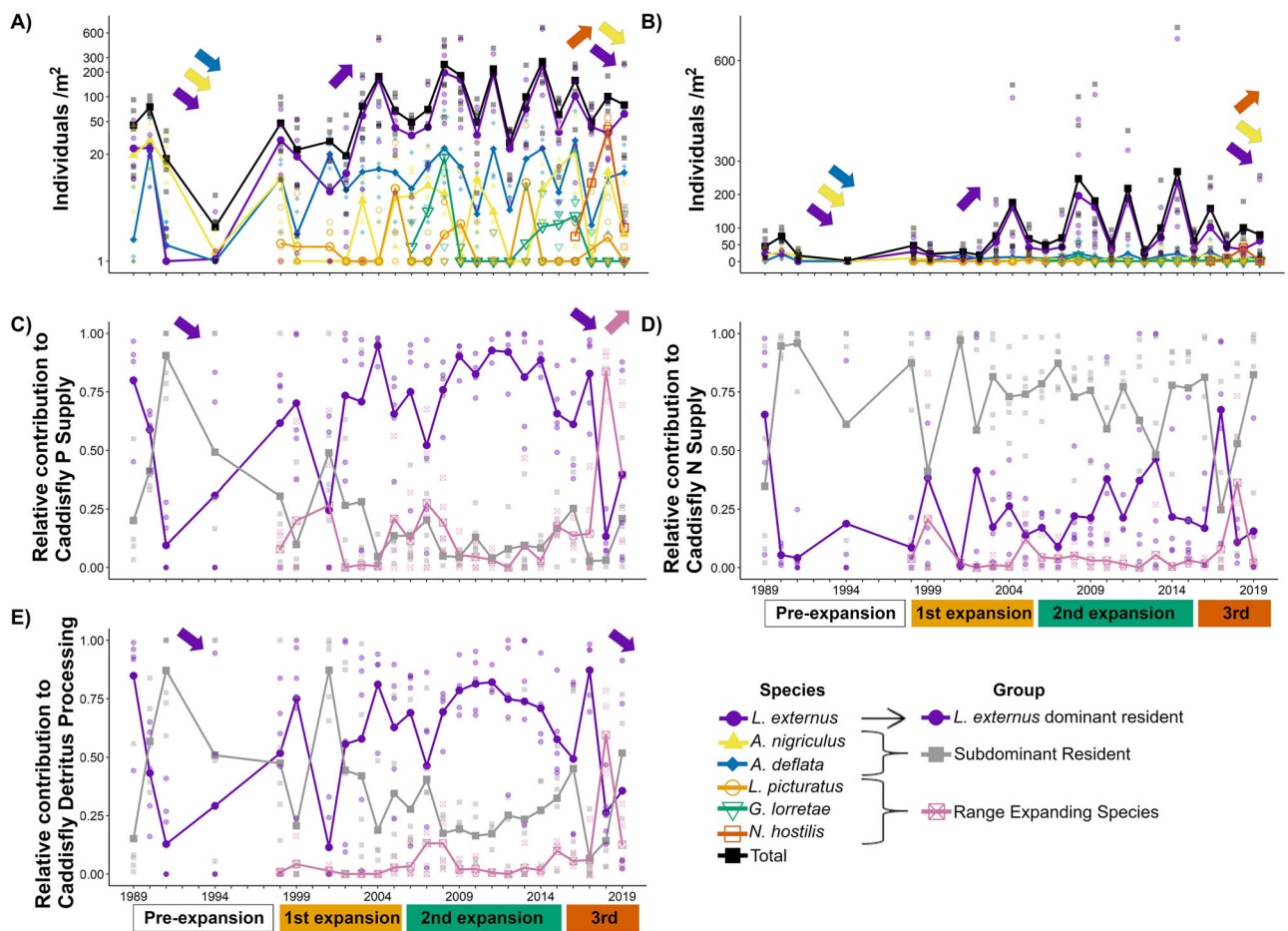

**Fig. 2 Larval caddisfly abundance and predicted relative contributions of caddisflies to ecosystem processes in permanent ponds over 30 years at Mexican Cut, Colorado.** Natural log-transformed caddisfly densities (**A**, note y-axis is backtransformed to original scale) were averaged across all permanent ponds (range = 3–7 ponds per year; individual pond means denoted by semitransparent points). Densities are also presented in original scale (**B**). Colored boxes under x-axes indicate range expansion periods defined by initial dates of upslope range expansion (e.g., orange first expansion box indicates *L. picturatus* arrived in 1998, green for *G. lorretae* in 2006, and red for *N. hostilis* in 2016). Species' relative contributions to the caddisfly assemblage's total were calculated to standardize for interannual variation in the total. We grouped the relative contributions to P supply (**C**), N supply (**D**), and CPOM processing (**E**) of subdominant residents (*Ag. deflata* and *An. nigriculus*) and range expanding species (*L. picturatus*, *G. lorretae*, *N. hostilis*) separately for comparison with the dominant resident (*L. externus*). Colored arrows within plot space indicate direction of statistically significant trends in corresponding species' abundance or groups' relative contribution during each range expansion.

caddisfly assemblage P supply and detritus processing throughout sequential range expansions. Conversely, during the third range expansion by *N. hostilis*, the dominant role of *L. externus* in regulating total P supply declined and the range expanding species provided redundancy towards maintaining ecosystem processes. Thus, even if range expansions did not alter these animals' total contributions to ecosystem processes, range expansions did reduce the relative functional role of the dominant resident while that of the range expanding species increased.

The extent to which functional roles of resident species can change following the arrival of range expanding species can be similar to changes from other drivers of population change. For example, prior to any caddisfly range expansions, populations of the dominant resident *L. externus* and subdominant residents *Agrypnia deflata* and *An. nigriculus* declined by 1.7, 1.3, and 1.6 individuals/m²/year (Fig. 2A; *L. externus* $p < 0.001$, *Ag. deflata* $p = 0.027$, *An. nigriculus* $p < 0.001$). Consequently, *L. externus*'s relative contributions to P supply and detritus processing declined by 9.0% annually (Fig. 2C, $p = 0.002$; 2E, $p < 0.001$), though the subdominant residents' relative contributions did not change (P supply: $p = 0.102$; detritus processing: $p = 0.133$). These population declines may be direct or indirect consequences

of consecutive early autumn freezes in 1990 and 1991 that occurred 17.5 days earlier (±1.6 standard deviations) than the 1989–2019 average. Such events would have killed *L. externus* adults during ovarian diapause or prevented successful oviposition by freezing pond surfaces[25,33]. Regardless of its cause, the annual decline in the relative contribution of *L. externus* following its population crash was similar to their contribution observed two decades later during *N. hostilis*'s range expansion (e.g., 9.0% vs 13.9%). Together, these events demonstrate that subdominant species may not show compensatory increases in abundance or functional roles following dominant species declines.

Range expansions could cause a wide range of outcomes for the functional roles of resident species. For example, range expansions can lead to local extinctions and replacement of residents[34]. Local extinction implies the complete loss of a species' contribution. Alternatively, facilitation effects or other changes in species interactions following range expansion could increase a resident's contribution. Although 'no change' was the most frequent outcome in our dataset, we suggest more studies are needed to determine if there is any generality in how species' functional roles or overall ecosystem processes are altered by

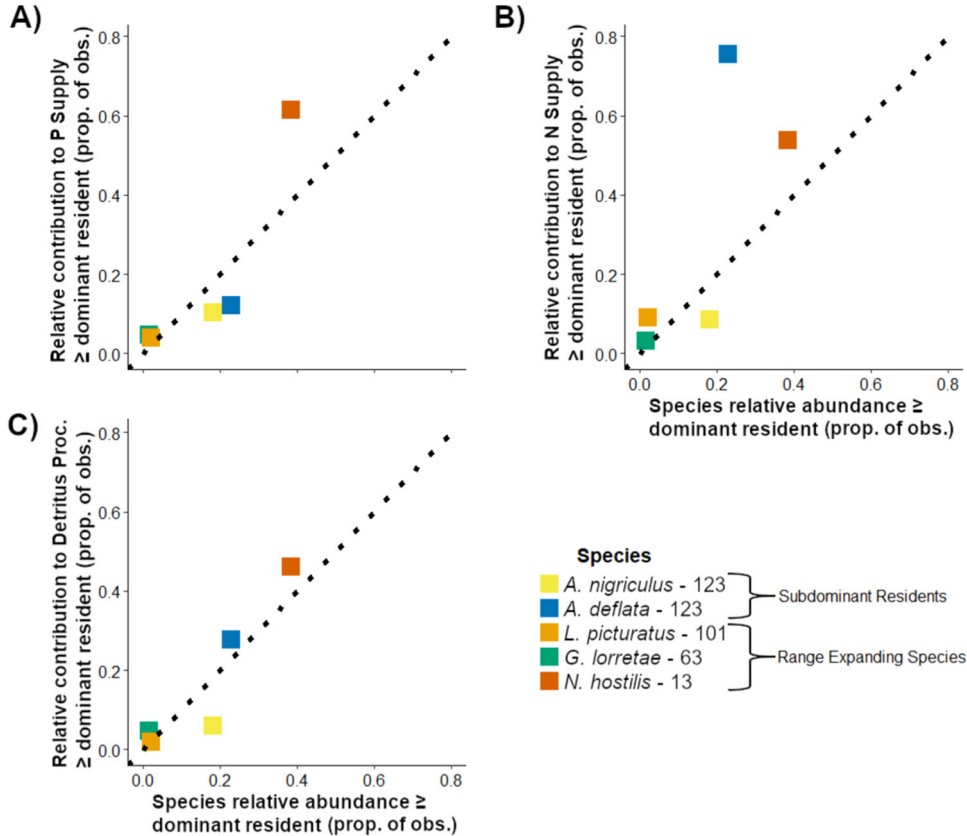

**Fig. 3 Proportions of pond-year observations where species' relative abundance and contribution to an ecosystem process equaled or exceeded those of dominant resident *L. externus*.** Numeric labels after species names in the legend indicate total number of ponds sampled since the species was first observed at Mexican Cut (e.g., total observations from all ponds across all years). Species above the dotted 1:1 lines can surpass the dominant resident's contribution to ecosystem processes without matching their relative abundance, whereas species below infrequently exceed the dominant resident's contribution despite greater abundance. Comparison of P supply contributions in (**A**), N supply in (**B**), and detritus processing in (**C**).

range expansions. Functional trait approaches could support such studies where total ecosystem processes are difficult to measure, or in communities where range expansions were previously documented but ecosystem processes were not measured.

**Traits influence residents' contributions and range expanding species success**. Uncommon species with unique functional traits can make contributions to ecosystem processes disproportionate to their abundance[35]. For example, subdominant resident *Ag. deflata* consistently contributed large proportions of N supply (~70.8%) throughout all three expansions (Fig. 2D). *Ag. deflata*'s large N contribution matched or exceeded that of dominant resident *L. externus* in 75.6% of pond-year observations despite only attaining equal or greater density in 22.8% (Fig. 3B). By extension, this suggests that range expanding species with low abundances and unique functional traits could also alter total ecosystem process rates (e.g., N or P supply or detritus processing by all species) or the relative contributions of a resident species. However, total ecosystem process rates and functional roles of resident species are likely more sensitive to the arrival of range expanding species that achieve high abundances because many organismal effects on ecosystem processes are driven by abundance or biomass rather than functional traits alone[15,36,37]. Consequently, the need to achieve and maintain high biomass suggests that range expanding species may be more likely to compete with and displace functional roles of dominant species that utilize more resources and habitat space than subdominants.

Thus, species life history and other traits that influence outcomes of competition with residents for resources or habitat

space could predict if range expanding species establish populations or alter ecosystem processes in recipient ecosystems[9,38]. Here, two upslope range expansions did not cause directional or even transient changes in species' relative functional roles, but one did. Specifically, during the most recent range expansion by *N. hostilis*, P supply contributions of the dominant resident *L. externus* declined annually while those of *N. hostilis* increased. We found that *N. hostilis* matched or exceeded *L. externus*'s relative abundance ~19.4x and ~24.2x more frequently than range expanding species *L. picturatus* and *G. lorretae*, though fewer ponds were surveyed since the third range expansion began (Fig. 3). Moreover, *N. hostilis* matched or exceeded *L. externus*'s ecosystem process contributions more frequently than the prior two range expansions combined by factors of 7.1x for P supply, 4.5x for N, and 6.9x for detritus processing. Thus, among the three range expanding species, *N. hostilis* appears to be the most likely to rise to numerical and functional prominence in permanent ponds. We suggest that *N. hostilis* could have achieved rapid population growth and functional displacement of the dominant resident due to its unique developmental phenology among caddisflies. Specifically, *N. hostilis* larvae hatch in the fall prior to pond freezing and complete initial stages of development when few other taxa are active, including salamander predators[18]. Consequently, early instar *N. hostilis* larvae in the fall likely encounter lower intraguild competition and predation pressure than spring-developing caddisfly taxa such as *L. externus* that are more vulnerable to both these interactions[26,39]. Furthermore, later onset of winter (Figure S2) could enhance this phenological advantage by

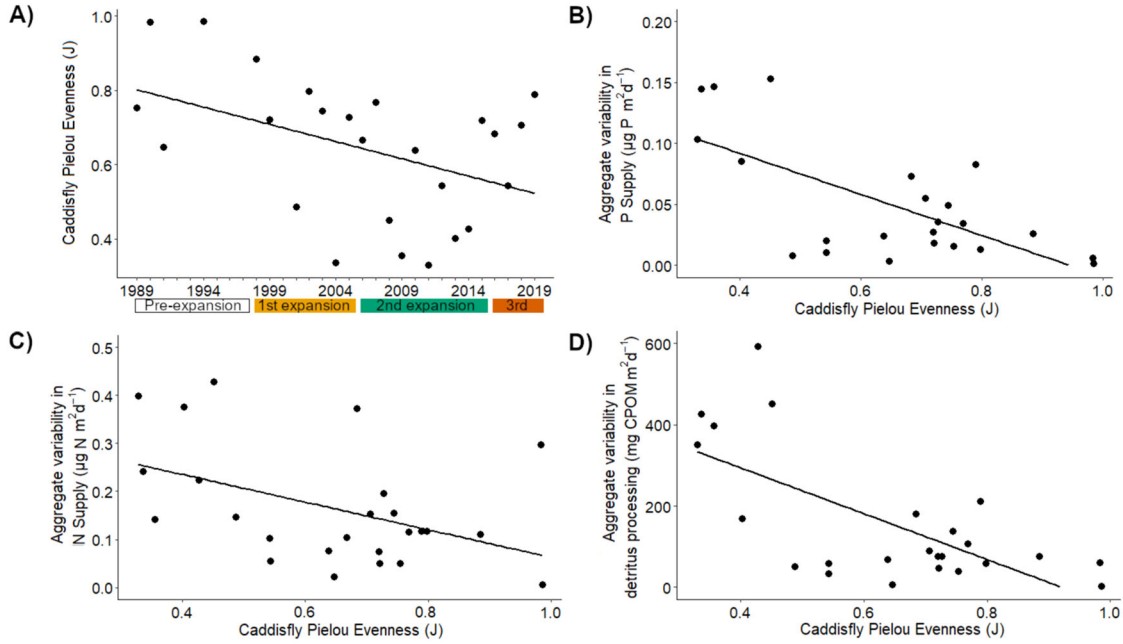

**Fig. 4 Relationships between caddisfly assemblage evenness and aggregate variability in species contributions to ecosystem processes.** Pielou's Evenness (J) was calculated using average caddisfly densities across all permanent ponds at Mexican Cut (e.g., densities in Fig. 2A; averages of = 3–7 ponds per year). Aggregate variability in species contributions to ecosystem processes (e.g., variability among ponds and species) was calculated as the sum of species' among-pond variance in process contribution and 2× the sum of contribution covariances among all species pairs. Aggregate variability was square root transformed to achieve normality for statistical analysis. This transformation also returns aggregate variability to the original units of the ecosystem process, corresponding to units of total contributions in Fig. S1A–C. Colored boxes under (**A**) x-axis indicate range expansion periods defined by initial dates of upslope range expansions (e.g., orange first range expansion box indicates *L. picturatus* arrived in 1998, green for *G. lorretae* and red for *N. hostilis* in 2016). Evenness declined over time ((**A**); $F_{1,17}$ = 5.73, $p$ = 0.029, $R^2$ = 0.20) but did not differ among or change during range expansions (expansion: $F_{3,17}$ = 1.74, $p$ = 0.20; expansion × year: $F_{3,17}$ = 0.72, $p$ = 0.55). All three ecosystem processes' aggregate variability had negative relationships with evenness (P in (**B**): $F_{1,23}$ = 17.61, $p$ < 0.001, $R^2$ = 0.42; N in (**C**): $F_{1,23}$ = 5.74, $p$ = 0.025, $R^2$ = 0.17; detritus in (**D**): $F_{1,23}$ = 17.83, $p$ < 0.001, $R^2$ = 0.43).

extending the window for the autumnal component of *N. hostilis* development. Finally, this key difference in *N. hostilis* life history could also extend the seasonal duration of caddisfly contributions to ecosystem processes beyond late summer when resident and all previous range expanding species pupate and overwinter in non-larval stages. Thus, unique life history traits (e.g., unique developmental phenology) may determine successful establishment in a new ecosystem[40], and likely also influence if and when range expanding species contribute to ecosystem processes and alter residents functional roles. Taken together, if the range expansion by *N. hostilis* is not transient, then this species could functionally replace the dominant resident, *L. externus*, and have the capacity to increase total ecosystem process rates. Indeed, *N. hostilis* has continued to increase in abundance in 2021 and 2022 (Greig and Balik *personal observations*).

**Range expansions and redundancy in ecosystem processes.** A more equal distribution of species abundances (high evenness) provides functional redundancy for ecosystem processes, whereas functioning is less resistant to environmental stressors in uneven assemblages dominated by a single species[28]. Our secondary objective was to explore this idea, with the expectation that evenness would increase with the addition of three new species and numerical declines in the dominant resident, resulting in a negative relationship between evenness and the aggregate variability of species' relative contributions to ecosystem processes. Aggregate variability in relative contributions includes variation in species' contributions among ponds (conceptually, among patches) and covariances among all species pairs[41,42]. Indeed, uneven caddisfly assemblages had more variable ecosystem process contributions, as aggregate variability in species'

contributions to all three ecosystem processes declined by 10.3x, 9.9x, and 3.7x from the least to most even caddisfly assemblage (Fig. 4). However, contrary to our expectation, caddisfly evenness declined over time (Fig. 4A), perhaps as a consequence of *L. externus* attaining greater numerical dominance following recovery from population decline in the early 1990s coupled with the arrival of range expanding species that attained low population sizes. Furthermore, the high aggregate variability in processes observed in these uneven assemblages (Fig. 4) exceeds the average of the total process contributions in some years (Figure S1), demonstrating that variation in the abundance of a dominant taxon among uneven assemblages can have very large effects on the assemblage's total contribution to ecosystem processes. By extension, if range expanding species do not sustain large populations and remain uncommon, their arrival may promote variability in ecosystem processes by reducing evenness, particularly in systems where historically dominant residents become increasingly abundant.

Finally, it is important to recognize that different taxonomic groups could be important for different ecosystem processes. Thus, for some ecosystem processes, contributions from non-caddisfly taxa could buffer against consequences of caddisfly range expansions. For example, in 2018 the caddisfly assemblage provided <2% of permanent ponds' total animal-driven N and P supply[23]. Instead, the bulk of animal-driven supply was provided by dipterans and zooplankton with higher mass-specific excretion rates. Although the caddisfly assemblage is biomass-dominant among benthic invertebrates[18], their low nutrient excretion rates may be constrained by the nutrient-poor coarse sedge detritus that comprises 78–95% of their diets[19], or by the need to rapidly attain large body sizes to consume more detritus[20], evade

predation[26,39], and complete larval development[25]. Indeed, low detritivore nutrient supply contributions are consistent with slower energy and material transfer to higher trophic levels through detrital pathways relative to autotrophic pathways[43]. In this larger community context, redundancy in animal-driven nutrient supply conferred by other invertebrate taxa would likely preclude any changes in total supply caused by arrival of range expanding caddisflies. However, other taxa are unlikely to provide comparable redundancy for multiple ecosystem processes, such as coarse detritus processing which is driven by larval caddisflies in this system[20,24]. Thus, ecosystem outcomes of range shifts and subsequent changes to animal or plant assemblages are likely to vary among ecosystem processes. Consequently, assessing the novelty of a focal assemblage's functional traits relative to those of other community members could help indicate which ecosystem processes are most likely to be sensitive to arrival of range expanding species.

**Conclusions**. In the broader context of range shifts, invasive species, or other compositional shifts that modulate ecosystem functioning, our results demonstrate that dominant and sub-dominant resident species can regulate ecosystem processes throughout sequential range expansions by species with similar life histories. In contrast, range expanding species with differing life histories can change relative contributions of species to ecosystem processes. Thus, in addition to functional traits that enable estimates of contributions to ecosystem processes, life history traits could be informative for predicting or interpreting functional consequences of compositional change, particularly when considered in tandem with long-term natural history observations.

## Methods

**Study sites and larval caddisfly natural history**. Ponds were located within the Mexican Cut Nature Preserve, a pristine, subalpine (3560 m) wilderness area owned by The Nature Conservancy and managed by the Rocky Mountain Biological Laboratory (RMBL) in the Elk Mountains of central Colorado. The Mexican Cut is a glacial cirque comprised of two shelves with 60+ kettle-pond wetland habitats, all with similar basin substrate composition and geomorphology, emergent and riparian vegetation, and water chemistry[18]. Ponds at the Mexican Cut are ecologically representative of high-elevation kettle-pond wetlands throughout the Rockies and other mountainous regions[44].

Annual censuses of pond communities since 1989 indicate that larval cased caddisflies (five species of Limnephlidae, one of Phryganeidae) dominate animal biomass, increasing from 30 to 70% of animal biomass from permanent to temporary ponds[18]. An additional Limnephilid caddisfly species, *Hesperophylax occidentalis*, is present at the Mexican Cut, but is restricted to the intermittent streams draining an alpine lake above the cirque's upper shelf. Larval caddisflies in lentic habitats were surveyed annually for 25/30 years between 1989 and 2019 with single 0.33 m² benthic D-net sweeps at the north, east, south, and west sides of 3–7 permanent, 2–11 semi-permanent, and 1–7 temporary ponds. Samples were collected between late June and mid-July each year, approximately 2–3 weeks after pond ice-out. This seasonal timing provides the best annual snapshot of the caddisfly assemblage given their staggered life histories. For example, four taxa overwinter as diapausing eggs that hatch when inundated with spring snowmelt. Among these, the temporary pond specialist *Asynarchus nigriculus* generally completes larval development and pupates before species adapted to permanent hydroperiods (*Limnephilus externus*, *L. picturatus*, *Grammotaulius lorretae*). Whereas, *Agrypinia deflata* hatch in the fall, overwinter as larvae, and complete development in mid-late summer around the same time as the Limnephilids. Finally, the most recent range expanding species, *Nemotaulius hostilis*, has a similar life history strategy as *Ag. deflata*, though it typically completes larval development earlier in the summer.

**Predicting caddis assemblage nutrient supply and detritus processing**. Previous work demonstrated that although there is high interspecific variation among the larval caddisflies' species-specific nutrient excretion, within a species excretion is strikingly consistent throughout the day and among pond habitats along an elevational gradient from montane to subalpine[19]. Furthermore, excretion declines predictably with developmental instar, though differences among species-specific instars are small relative to interspecific differences. In addition, there is comparable interspecific variation in detritus processing rates measured in laboratory

microcosms over the course of larval development[20], and species' microcosm processing rates are generally within 20% of species-specific processing rates measured in-situ with littoral cages[27,45]. Furthermore, this previous work demonstrates that additive predictions of caddisfly assemblage detritus processing calculated by summing the products of species' mass-specific processing rate and biomass provide accurate estimates of in situ assemblage-total detritus processing[20]. Thus, we predicted caddisfly species' contribution to ecosystem processes over time as the products of average density, final instar mass, larval development rate, and either nutrient excretion rate or detritus processing rate.

To incorporate variation in species-specific detritus processing and nutrient excretion rates into our additive predictions of caddisfly contributions to ecosystem processes, we adapted a random sampling framework from ref. [23]. Specifically, pond-level calculations of species' nutrient supply and detritus processing contributions across the long-term survey dataset were repeated 1000 times with randomly sampled species-specific rates in each iteration. Rather than estimating the caddisfly assemblage's contribution to ecosystem processes for the empirical survey ponds, we instead use our pond-year caddisfly assemblage densities to estimate their contributions in an average permanent pond following the methods of ref. [23]. This approach is advantageous for three reasons. First, larval caddisflies tend to congregate in the shallow littoral habitats of pond perimeters near emergent sedge vegetation and are not distributed evenly across deeper pond centers. Thus, our census samples collected near pond shorelines likely over-estimate caddisfly population density in an average square meter of pond benthos. The random sampling framework corrects for this by simulating a random pond for each empirical pond-year caddis assemblage. Specifically, random samples from hydroperiod-specific normal distributions of pond area and % of pond area habitable to larval caddisflies are used to rescale empirical caddisfly assemblage densities to a square meter of total pond area. Second, we do not have data describing total pond area or area habitable to larval caddisflies for all survey ponds. Because we cannot appropriately rescale caddisfly population densities for all empirical ponds, simulating a random pond for each pond-year caddisfly assemblage 1000 times allows us to utilize all pond-year survey samples to predict the caddisfly assemblage's average contributions to ecosystem processes. Likewise, third, although some permanent ponds were surveyed annually (n = 2 of 7), others were surveyed opportunistically. The simulation framework thus leverages all pond-year surveys to estimate the caddisfly assemblage's average contributions.

Predicted species-specific ecosystem process contributions for each pond-year were averaged across all iterations to estimate species' contributions in an average permanent pond. Thus, among-pond variation in species-specific predicted contributions is due to variation in species' density, standardized for known variation in pond size and area habitable to larval caddisflies. Species-specific contributions were then summed within pond-years to estimate the caddisfly assemblage's total contribution. To compare species' contributions in subsequent statistical analyses and standardize for interannual variation in the assemblage-total, species' relative contributions to the assemblage-totals were calculated. Finally, we grouped subdominant species' and range expanding species' relative contributions separately for comparison with the presumed dominant resident *L. externus*[25,35].

Species average pond-specific contributions (e.g., species average contribution across all simulation iterations within a given pond) were used to estimate aggregate variability of caddisflies contributions to ecosystem processes among ponds following methods of refs. [41,42]. Specifically, each species' among-pond variance in process contributions were summed and added to 2x the sum of covariances among all species pairs within each pond-year. We then square-root transformed aggregate variability to improve normality for statistical analysis. Notably, this transformation also returns aggregate variability to the original data's scale, as untransformed aggregate variability shares the squared units of its variance and covariance components.

**Statistical analyses**. To address our primary objective of determining if species groups' relative contributions differed among or changed during time periods that correspond with upslope range expansions, we created a categorical variable called "Range Expansion" to group years between sequential upslope range expansions. The category "pre-range expansion" was applied to 1997 and earlier, whereas years between 1998–2005 were categorized as "1st expansion", 2006–2015 as "2nd expansion", and 2016–2019 as "3rd expansion".

All statistical analyses were completed in R 4.0.2[46]. The package "lme4" was used to fit mixed effects models to caddisfly relative abundances, assemblage-total predicted ecosystem processes, and species' relative contributions over time[47]. For mixed models with caddisflies total predicted ecosystem process as the response variable, Year and Range Expansion were modeled as fixed effects along with their interaction. Whereas, when relative contributions or abundance were the response variable, Year, Range Expansion, Group (for relative contributions, i.e., dominant resident, subdominant residents, range expanding species) or Species (for relative abundance), and their interactions were modeled as fixed effects. All mixed models also included Pond as a random effect, and a third-order autocorrelation function (ACF) of year nested within pond. Inclusion of an appropriate ACF order was determined by examining ACF plots for dominant taxon relative abundance; the ACF function for *L. externus* relative abundance exceeded the α = 0.05 confidence region at lag = 3.

Following mixed effects models of species' abundances or relative contributions with significant interactions (e.g., Year*Range Expansion*Species, Year*Range Expansion*Group, Year*Species, or Year*Group) the R package "emmeans"[48] was used to compare the slope estimates of each Group's relative contribution over time (Year) within each range expansion. Slope estimates were compared against a null hypothesis of slope = 0 using linear contrasts.

Finally, for our second objective, we compared aggregate variability to caddisfly evenness (J, Pielou) with simple linear models (model: square-root transformed aggregate variability in N or P supply or detritus processing = evenness).

**Reporting summary**. Further information on research design is available in the Nature Portfolio Reporting Summary linked to this article.

## Data availability

All data included in an electronic supplement (Supplementary Data 1).

## Code availability

Novel code for predicting caddis assemblage nutrient supply and detritus processing included in electronic supplement (Supplementary Data 1).

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

## Acknowledgements

We thank the Rocky Mountain Biological Laboratory for logistical support and The Nature Conservancy for access to the Mexican Cut Nature Preserve. We also thank W.S. Brown, undergraduate students from Allegheny College, and A.J. Klemmer for assistance with data collection. We are grateful to L.M. Demi, R.R. Dunn, A.J. Klemmer, S.E. Jordt, I.D. Shepard, S.E. Washko, B.L. Peckarsky, M.T. Bogan, and three anonymous reviewers for constructive discussions and suggestions that greatly improved the manuscript. Finally, we thank Allegheny College, R.L. Mumme, M.D. Venesky, C. Wilson, E.L. Coates, and most importantly Sue Wissinger for securing and permitting access to S.A. Wissinger's data in January 2020. This work was supported by National Science Foundation grants 1556914 and 1641041 to B.W.T., 1557015 to S.A.W., and 1556788 to H.S.G. H.S.G. was also supported by the USDA National Institute of Food and Agriculture, Hatch project number #ME0-22112 through the Maine Agricultural & Forest Experiment Station. J.A.B. was supported by an NSF GRFP and an NCSU Provost Graduate Student Fellowship.

## Author contributions

All coauthors conceptualized this research and contributed to the long-term caddisfly survey efforts led by S.A.W. J.A.B. analyzed data, produced visuals, and wrote the manuscript with substantial intellectual contributions and feedback from B.W.T. and H.S.G.

## Competing interests

The authors declare no competing interests.
