## [Peer Review File · Communications Biology]

Reviewers' comments:

Reviewer #1 (Remarks to the Author):

Balik et al. used a long-term dataset on caddisfly abundances in a mountain pond system to test predictions about how species range shifts and associated changes in community composition can alter key freshwater ecosystem processes. This is a highly relevant topic in the context of climate change, and the approach used here is excellent. They make use of existing data – including caddisfly abundances in ponds for a 30-yr period and estimates of species-specific rates of N and P excretion/supply and detritus processing – to demonstrate that species range expansions can alter the relative contributions of different species to key ecosystem processes, even when total process rates remain unaltered. I found that both the study and manuscript (figures included) are flawless and thus have no single criticism (first time ever!) 'Limnephilus' is misspelled in some occasions.

Reviewer #2 (Remarks to the Author):

The manuscript "Consequences of climate-induced range expansions on multiple ecosystem functions" by Balik et al., presents a very thorough analysis of decades of data indicating ecosystem changes upon species range expansions. The analysis focuses on one important macroinvertebrate group, however, the data analysis is well-conceived in a scheme that testing fixed and random factors provides a clear contribution of caddisfly to different ecosystem processes. The selection of testing evenness is also an insightful metric that shows the role of functional traits and abundance to cope with changes in range expansions. I comment Balik et al., for this well-presented analysis that tackles main ecosystems processes and climate change from a range expansions novel perspective.

Regards,

Daniela Rosero-Lopez

Jared A. Balik, Postdoctoral Scholar
North Carolina State University
Department of Applied Ecology
Raleigh, NC 27695
724.683.9774
Balikj3@gmail.com

Response to referee's comments for Communications Biology manuscript COMMSBIO-22-3451-T, "Consequences of climate-induced range expansions on multiple ecosystem functions". Here, reviewer comments are in plain black text, whereas our responses are in blue bolded italics.

Reviewers' comments:

Reviewer #1 (Remarks to the Author):

Balik et al. used a long-term dataset on caddisfly abundances in a mountain pond system to test predictions about how species range shifts and associated changes in community composition can alter key freshwater ecosystem processes. This is a highly relevant topic in the context of climate change, and the approach used here is excellent. They make use of existing data – including caddisfly abundances in ponds for a 30-yr period and estimates of species-specific rates of N and P excretion/supply and detritus processing – to demonstrate that species range expansions can alter the relative contributions of different species to key ecosystem processes, even when total process rates remain unaltered. I found that both the study and manuscript (figures included) are flawless and thus have no single criticism (first time ever!) 'Limnophilus' is misspelled in some occasions.

Thank you! We have corrected the spelling of 'Limnophilus' at lines 59, 80, and 115 (highlighted; now lines 59, 80, 118).

Reviewer #2 (Remarks to the Author):

The manuscript "Consequences of climate-induced range expansions on multiple ecosystem functions" by Balik et al., presents a very thorough analysis of decades of data indicating ecosystem changes upon species range expansions. The analysis focuses on one important macroinvertebrate group, however, the data analysis is well-conceived in a scheme that testing fixed and random factors provides a clear contribution of caddisfly to different ecosystem processes. The selection of testing evenness is also an insightful metric that shows the role of functional traits and abundance to cope with changes in range expansions. I comment Balik et al., for this well-presented analysis that tackles main ecosystems processes and climate change from a range expansions novel perspective.

Regards,
Daniela Rosero-Lopez

Thank you!

Line 81 - There seems to be several expected outcomes. I suggest classify them to show the "primary hypothesis" as stated in line 215"

Excellent point, we agree this framing would help clarify our overall objectives for first time readers. We have modified a sentence at line 76 to indicate that our primary objective was to explore how the predicted relative contributions to multiple ecosystem processes of a dominant resident, a group of subdominant residents, and a group of range expanding species changed over time throughout three

Jared A. Balik, Postdoctoral Scholar
North Carolina State University
Department of Applied Ecology
Raleigh, NC 27695
724.683.9774
Balikj3@gmail.com

*range expansions. The following unchanged sentences from lines 79-85 then detail the two hypotheses that 1) successive range expansions would reduce the abundance of dominant resident *Limnephilus externus*, causing declines in its historically large relative contribution to the caddisfly assemblage's total nutrient supply and detritus processing, and 2) abundance of range expanding species would increase over time, along with their relative contributions to ecosystem processes.*

Next, at line 85, we have added a brief sentence stating that our second objective was to explore how any changes in caddisfly assemblage evenness resulting from successive range expansions would influence redundancy in the assemblage's relative contributions to ecosystem processes.

Subsequently, these framings are added in the 'Statistical analysis' methods section at lines 183 and 206 to clarify what statistical approaches were used for each objective.

Line 170 – do you need a “to” here?

Yes, we have added it. Thanks for catching this. (Now at line 173)

Line 194 – do you mean “an”?

Yes, we have added it. (Now at line 198)

Line 215 – Read comment on line 81

Here we made minor edits to reiterate the “primary objective” framing used above. This is now at line 219. At line 342 we made a brief edit to reiterate the “secondary objective”.